# Algorithmic Complexity Predicts when Path Information Improves Graph Neural Networks Performance on Molecular Graphs.

## Abstract

Graph Neural Networks (GNNs) are designed to process irregular relational data in recommendation systems, protein networks, social networks, and molecules. GNNs typically rely on message passing and aggregation, with some architectures incorporating graph path information in a bid to improve accuracy. However, it is unclear whether such incorporation of path information truly improves GNN accuracy in all cases. As a first step, we herein shed light on this issue for the case of molecular graphs. We evaluated Graphormer and Mix-Hop models, with and without path information on 36 molecular datasets, derived from six MoleculeNet benchmark datasets. Path information improved performance in some cases but not in other cases. This finding is important, because these two models always incorporate path information in practice, whereas the finding shows this incorporation of path information can actually be detrimental to the models' accuracies. To more deeply probe this observation, we developed a graph representation model called T-hop which allows us to further highlight the use, versus non-use, of path information. On one hand, we formulate the Path Usefulness Measure (PUM) to quantify the benefit of path information. On the other hand, we quantified the randomness of the different datasets via their algorithmic complexities, using the Block Decomposition Method (BDM). We hypothesized, and confirmed our hypothesis, that: GNN models trained on molecular datasets with less random structures (i.e. lower algorithmic complexity) should benefit from path information (i.e. larger PUM), compared to datasets with more random structures. In summary, low algorithmic complexity, which captures the presence of structure in molecular graphs, is useful for predicting when path information improves accuracies in GNNs. A practical benefit of this is that it leads to a more resource-efficient approach, wherein path information is only incorporated for datasets with low algorithmic complexities.

## 1 Introduction

Graphs are an important category of data in machine learning because many interesting entities and phenomena, such as text, brain connectivity networks, molecules, protein networks and social networks, can be naturally represented as graphs. The most prominent class of models for processing graphs in machine learning comprises graph neural networks (GNN), perhaps due to their success on real-world datasets (Shen et al., 2025; Wang et al., 2025; Yu et al., 2024; Luan et al., 2021; Kipf & Welling, 2017). Most GNNs follow a message passing paradigm (Gilmer et al., 2017) (Yang et al., 2019), which involves a three-stage process. In the first stage, the messages to be passed are computed. In the second stage, the messages are aggregated according to the connectivity constraints of the graph. Finally, in the third stage, the aggregated messages are used to update the appropriate node features. Furthermore, some GNNs, such as graphormer (Ying et al., 2021) and Mix-Hop (Abu-El-Haija et al., 2019), incorporate path information, in a bid to improve accuracy. In these models, the underlying assumption is that incorporation of graph path information should always improve models' accuracies. However, as we shall show experimentally in this work, this assumption can be wrong in some cases. In particular, our experimental results show that path information improves

accuracies in some cases, while lowering accuracies in other cases. An important question to ask then is: under what conditions does path information yield the desired gains in GNN accuracies, and thereby justifies the additional computational expenses concomitant with computing the path information ?

To the best of our knowledge, there is no existing work in the literature that characterizes conditions under which path information is expected to improve GNN accuracies. This is the case, albeit a plethora of GNN models that incorporate path information in the same literature (Ying et al., 2021; Abu-El-Haija et al., 2019; Li et al., 2019; Jin et al., 2021). For example, the popular graphormer model (Ying et al., 2021) incorporates path information by utilizing information along the shortest path between every pair of nodes in the input graph and augmenting its attention matrices with that information. Another GNN model that incorporates path information is the Mix-Hop model (Abu-El-Haija et al., 2019). It does so by utilizing a set of powered adjacency matrices whose entries encode path information. Yet another model that uses path information, although to a lesser extent than Mix-Hop, is the IGCN (Improved Graph Convolutional Network) model (Li et al., 2019). IGCN uses a powered symmetrically-normalised identity-shifted adjacency matrix, which can ultimately be expressed in terms of the powered adjacency matrix. Furthermore, similarly to Mix-Hop, Power-Up (Jin et al., 2021) is also a GNN model that utilizes path information. It does so by using a set of matrices that encode information about the path length of the shortest distance between all pairs of nodes in the graph in question. In addition, to incorporate path information, a line of work (Kong et al., 2022; Sun et al., 2022; Michel et al., 2023; Neelakantan et al., 2015), explicitly considers the sequence of paths leading to the nodes of a graph. Other works incorporating path information include NBFNet (Neural Bellman-Ford Networks) (Zhu et al., 2021) and methods that can be considered as its special cases (Katz, 1953; Liben-Nowell & Kleinberg, 2007; Page et al., 1999).

Some earlier works already indicate that adding path information to GNNs does not always improve accuracy (Li et al., 2019; Michel et al., 2023; Cong et al., 2023). For example, in the PathNN method (Michel et al., 2023), while increasing path length improved results in the NCI1 data set of the TUDataset collection (Morris et al., 2020), it failed to improve results using the PROTEINS and ENZYMES data sets of the same TUDataset collection. Likewise, in the IGCN paper (Li et al., 2019), experiments using the AWA2 dataset Xian et al. (2018) showed worse performance as $k$ increased from 1 to 3, where $k$ can be viewed as a parameter that controls the maximum path length incorporated into the IGCN model. Moreover, even for the more recent GraphMixer Cong et al. (2023), it was found that using a larger receptive field, and hence path length, diminished accuracies. However, these earlier works neither asked nor answered the question of under which conditions graph path information can be expected to increase GNN accuracies.

In this work, we take the first steps towards addressing the above question for the case of molecular graphs. We first examined two widely used GNNs: Graphormer and Mix-Hop. For each model, we designed two modes: a first mode that jettisons path information altogether and a second mode that utilizes path information. We then compared the two modes on six MoleculeNet graph datasets Wu et al. (2018). We found that path information did not consistently improve performance in these models. This finding is consequential, because these two models always incorporate path information in practice, whereas this finding shows this can actually be detrimental to their accuracies. To study this phenomenon more closely, we designed a novel model, which we call T-Hop, which more aggressively amplifies the dichotomy between the use of path information on the one hand and its non-use on the other hand. Again using path information with the T-Hop model also did not improve accuracy in all cases. This prompted us to search for a deeper understanding of the reason behind these counter-intuitive results. We conjectured that the extent to which a graph dataset benefits from path information in GNN models might be inversely correlated with the level of randomness in the dataset's graphs' path connectivity patterns. To test this conjecture, we created six dataset families. Each data set family comprises one of six original MoleculeNet datasets along with five synthetic variants of the original data set. On one hand, we quantified the randomness in the dataset families' graph structures via algorithmic complexity (Zenil et al., 2023; 2018b), using the BDM (Zenil et al., 2023; 2018a). On the other hand, we quantified the benefit of path information in the GNN models via the PUM score, which we introduce later in this work. We found that GNN models trained on dataset families with lower algorithmic complexities benefited more from the incorporation of path information and hence have higher PUMs, as opposed to those trained on families with higher algorithmic complexities. We underscored this observation by showcasing strong negative Pearson correlations between algorithmic complexities and the PUMs, thereby

corroborating our conjecture. This was further consolidated by our clustering analysis, which showed strong matches between the dataset clusters created using the PUMs on the one hand, and those created using the algorithmic complexities on the other hand.

## 2 Methods

In this section, we outline our methodology. In Section 2.1, we describe the molecular datasets employed in this work. In Section 2.2, we give a brief description of how algorithmic complexity (Zenil et al., 2023; 2018b), implemented via the BDM (Zenil et al., 2023; 2018a), can be used to quantify the level of randomness/irregularities within the path connectivity patterns of graphs. In Section 2.3, we describe the three models used in this work. Finally, in Section 2.4, we introduce the PUM as a score for quantifying the usefulness of path information in GNNs.

### 2.1 Datasets

For our numerical experiments, we used a total of thirty six molecular datasets – six original non-synthetic MoleculeNet datasets (Wu et al., 2018) along with thirty additional synthetic datasets derived from the original six datasets. MoleculeNet is a suite of benchmark molecular datasets often used to calibrate GNN models. Each dataset in MoleculeNet comprises a list of molecules, represented via SMILES (Simplified Molecular Input Line Entry System) (Weininger, 1988), along with corresponding properties of interest. Examples of properties of interest are water solubility values, blood-brain permeability, clinical trial toxicity, etc. In this work, we used the following six original non-synthetic MoleculeNet datasets: FreeSolv, ESOL, Lipophilicity, BACE, BBBP and ClinTox (Wu et al., 2018). Table 1 summarizes pertinent information about these six datasets.

Table 1: Description of datasets

| DATASET | TOTAL NO. OF GRAPHS | TASK | TASK TYPE | METRIC |
|---|---|---|---|---|
| FreeSolv | 643 | hydration free energy | regression | RMSE |
| ESOL | 1128 | solubility in water | regression | RMSE |
| Lipophilicity | 4200 | octanol/water distribution coeff. | regression | RMSE |
| BACE | 1522 | $\beta$-secretase binding | classification | ROC-AUC |
| BBBP | 2053 | blood-brain barrier crossing | classification | ROC-AUC |
| ClinTox | 1491 | clinical trial toxicity and FDA approval | classification | ROC-AUC |

For processing the graphs in our molecular datasets, we harnessed the in-built software functionalities in the Deep Graph Library (DGL) (Wang et al., 2019) and DGL-LifeSci (Li et al., 2021). DGL-LifeSci includes six MoleculeNet datasets relevant to this work. Furthermore, it gave us a convenient workflow for converting the SMILES representation used in the MoleculeNet datasets to DGL graph representations, which are based on a more intuitive list of edges graph representation (LOEGR). In the LOEGR, the nodes of the graph correspond to the atoms of the underlying molecule, while the edges of the graph correspond to the bonds

between the atoms. This conversion is illustrated in Figure 1 for the case of methanoic acid, the simplest known organic acid in nature. As illustrated by the top-most matrix in the figure, DGL's LOEGR of a graph is implemented as a Pytorch 2-d matrix, wherein each column of the 2-d matrix represents an edge of the graph. Furthermore, the figure also shows that the duo of DGL-LifeSci and DGL makes it possible to easily compute

node and edge features for the graph. These are respectively represented by the middle and bottom matrices in the figure. Some models, such as Graphormer, require both the node and edge features along with the graph's LOEGR, while other models, such as Mix-Hop and T-Hop, require only the node features along with the graph's LOEGR as input. The fact that DGL's graph representations are implemented as Pytorch matrices makes them especially suitable as inputs to the Pytorch implementations of our models, which will be discussed in the coming sub-sections.

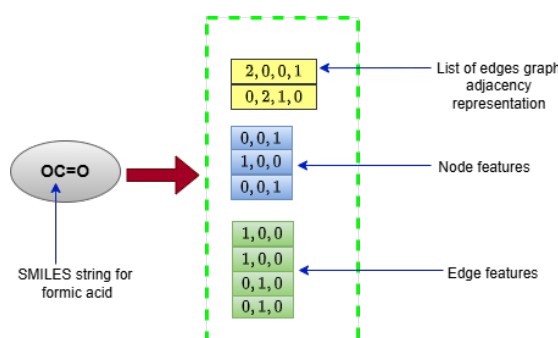

Figure 1: DGL/DGL-LifeSci conversion of methanoic acid's SMILES representation to a graph representation comprising: 1). a list of edges representation using a Pytorch 2-d matrix with each column representing an edge (top matrix in the figure); 2) a Pytorch 2-d matrix containing the graph's nodes' features (middle matrix in the figure ); 3) a Pytorch 2-d matrix containing the graph's edges' features (bottom matrix in figure )

To suit the aims of our experiments, we required a collection of datasets that share the same path connectivity information, but with different node and edge features. To this end, we defined *dataset families*. A dataset family comprises an original non-synthetic graph dataset, along with a collection of synthetic graph datasets derived from the original dataset. This derivation is done via the addition of pre-defined levels of Gaussian noise to the node and edge features of the graphs in the original non-synthetic dataset, without mutating the path connectivity patterns within those graphs. Note that this means that **datasets within the same dataset family share the same set of path connectivity patterns**. To create the dataset families used in this work, we started with the six original non-synthetic MoleculeNet datasets mentioned in the preceding paragraph. We then introduced a noise-level parameter, $\gamma$, $\gamma \in \{0.0, 0.1, 0.2, 0.3, 0.4, 0.5\}$. Also, $\sigma_0$ denotes the standard deviation of an original non-synthetic dataset's features. Then for each original dataset, we computed the set, $S_{\sigma_0} = \{0.0\sigma_0, 0.1\sigma_0, 0.2\sigma_0, 0.3\sigma_0, 0.4\sigma_0, 0.5\sigma_0\}$. To each original dataset, we then added zero-mean Gaussian noises with standard deviations $\sigma \in S_{\sigma_0}$ to the dataset's features. This yielded five synthetic variants of the original dataset. Here, note that $\sigma = 0.0\sigma_0$ is the noise-less case, which actually corresponds to the original non-synthetic dataset. In all, the above procedure allowed us to obtain a total of thirty six datasets: thirty synthetic datasets along with the six original non-synthetic datasets. For our experiments on each of the thirty six resulting datasets, we used scaffold splitting (Wu et al., 2018) to split the dataset into training:validation:test segments in the ratio 80:10:10. Finally, we point out that each of the 30 synthetic datasets inherit properties listed in Table 1 for its parent non-synthetic dataset.

## 2.2 Algorithmic Complexity and BDM

We required a way to measure the randomness (i.e lack of regularity) of the connectivity patterns within the graphs in our datasets. We used algorithmic complexity (Zenil et al., 2023; 2018b) as a measure of this randomness, and employed the BDM (Zenil et al., 2023; 2018a) for approximating it. We favored algorithmic complexity over traditional measures of randomness, such as Shanon entropy, because the former has been demonstrated to be better at capturing the intrinsic information content of a graph's structure, beyond sheer statistical patterns (Zenil et al., 2023). In BDM (Zenil et al., 2023; 2018a), the adjacency matrix, $A$, of the input graph, $G$, is first partitioned into a set of non-overlapping $d \times d$ sub-arrays. The $j$-th sub-array is denoted $r_j$. Then, the Coding Theorem Method (CTM) (Zenil et al., 2023) is applied, to compute the algorithmic complexity, $K_{CTM}(r_j)$, of $r_j$. This involves computing the algorithmic probability of $r_j$. To compute the algorithmic probability, a large collection of 2-d Turing machines (i.e. turmites) is run. The

frequency with which $r_j$ is output by the collection is then computed, and used to approximate $K_{CTM}(r_j)$. In symbols, the BDM algorithmic complexity of $G$, denoted $K_{BDM}(G, d)$, is expressed in terms of $K_{CTM}(r_j)$ as:

$$K_{BDM}(G, d) = \sum_{(r_j, n_j) \in A} K_{CTM}(r_j) + \log_2(n_j) \tag{1}$$

Above, $n_j$ is the number of times $r_j$ occurs within $A$. The notation $\sum_{(r_j, n_j) \in A}$, as opposed to $\sum_{r_j \in A}$, serves to emphasize that we count $K_{CTM}(r_j)$ separately for each of the $n_j$ occurrences of $r_j$ within $A$. To eschew re-inventing the wheel, we leveraged the publicly available implementation of BDM at the URL:
https://pybdm-docs.readthedocs.io/en/latest/pybdm.html

### 2.3 Models

This section describes the three models employed in this study, namely Graphormer (Ying et al., 2021), Mix-Hop (Abu-El-Haija et al., 2019) and T-Hop. The T-Hop model is tensor-based graph representation model we introduce in this work. We introduce it as a more effective model for highlighting the dichotomy between the use of path information versus the non-use of path information in GNNs.

#### 2.3.1 Synopsis of the Graphormer model

Here, we describe how to use the graphormer model for the goals of this work. For a full account of the graphormer model, the interested reader is referred to the graphormer paper (Ying et al., 2021). Essentially, Graphormer can be viewed as an analog of the popular transformer model (Vaswani et al., 2017) applied to graph datasets. At each layer of graphormer, the transformer-style softmax attention mechanism is applied to every pair of nodes in the input graph; followed by a skip-connection and the LayerNorm; followed by a feedforward MLP; followed again by a skip-connection and LayerNorm. In addition, the graphormer model super-imposes the structure of the underlying graph into the model by augmenting the softmax attention matrix with path information. The path information here is derived by considering the distance along the shortest path between every pair of nodes, as well as by considering the flow of edge features along those shortest paths. Consequently, for our experiments in this paper, we implemented two modes of the graphormer model: a first mode, wherein the softmax attention matrix is not augmented with any path information; and a second mode wherein the softmax attention matrix is augmented with path information as above described. Our Pytorch implementation can be found at the URL:
https://github.com/rahmanoladi/kaust_path_project/tree/main/path_project/graphormer

#### 2.3.2 Synopsis of the Mix-Hop model

The Mix-Hop model (Abu-El-Haija et al., 2019) harnesses a collection of powered adjacency matrices, $\mathcal{S} = \{A, A^2, \ldots, A^{L_m}\}$, with $L_m$ being the maximum path length considered in the model. Specifically, where $\|_{L=1}^{L_m}$ denotes the vector concatenation operation; where $H^l$ denotes input feature matrix to the $l$-th Mix-Hop layer; where $W_L^l$ denotes a learnable matrix; and where $\sigma$ denotes an activation function, then each layer of Mix-Hop is underpinned by the following formula:

$$H^{l+1} = \Big\|_{L=1}^{L_m} \sigma(A^L H^l W_L^l) \tag{2}$$

Since $L_m$ denotes maximum path length in the above model, it follows that we can implement a mode that does not incorporate any path information into the model by setting $L_m = 1$. On the other hand, we can implement a mode that incorporates path information into the model by setting $L_m > 1$. For our experiments, we implemented these two modes in Pytorch. Our implementation can be found at:
https://github.com/rahmanoladi/kaust_path_project/tree/main/path_project/mix_hop

### 2.3.3   The T-Hop model: A novel tensor-based approach

This section introduces T-Hop, a model that sharply contrasts performance with and without path information. Let $G = (V, E)$ represent a graph, where $V = \{v_1, ..., v_n\}$ and $E = \{e_1, ..., e_m\}$ are the nodes and edges of $G$, as usual. The adjacency matrix of $G$ is $A$. We consider two arbitrary nodes, $v_i$ and $v_j$ in $G$. Let $L \geq 2$, and let $\mathcal{B}_{i,j,k}^L$ be the number of paths of length $L$ between $v_i$ and $v_j$ that contain $v_k$. We may arrange the values, $\mathcal{B}_{i,j,k}^L$, in an $n \times n \times n$ 3-d tensor denoted $\mathcal{B}^L \in \mathbb{R}^{n \times n \times n}$, such that the entry on the $i$-th row, $j$-th column and $k$-th depth of $\mathcal{B}^L$ is $\mathcal{B}_{i,j,k}^L$. Note that the tensor, $\mathcal{B}^L$, encapsulates path information contained in the input graph, $G$. For theoretical reasons, we wish to scale $\mathcal{B}^L$ by $L+1$, and denote the result as a new tensor, $\mathcal{T}^L \in \mathbb{R}^{n \times n \times n}$:

$$\mathcal{T}_{i,j,k}^L = \frac{\mathcal{B}_{i,j,k}^L}{(L+1)} \tag{3}$$

Note that, like the tensor, $\mathcal{B}^L$, the tensor $\mathcal{T}^L$ also encapsulates path information contained in the input graph, $G$. To proceed, we define a new $n \times n$ matrix $\mathcal{M}$ as follows:

$$\mathcal{M} = \alpha_0 A + \sum_{L=2}^{L_m} \sum_{k=0}^{n-1} \alpha_{L,k} \mathcal{T}_{:,:,k}^L \tag{4}$$

Above, $A$ is adjacency matrix, while $\alpha_0$ and $\alpha_{L,k}$ are learnable scalars. Further, $L_m$ is the maximum path length considered in the model. Also, $\mathcal{T}_{:,:,k}^L \in \mathbb{R}^{n \times n}$ is the 2-d matrix slice associated with depth-$k$ of $\mathcal{T}^L$. Using $\mathcal{M}$, the $l$-th layer of our T-Hop model can now be described as:

$$H^{l+1} = \sigma(\mathcal{M} H^l W^l) \tag{5}$$

Above, $H^l$ denotes input features to the $l$-th layer, $W^l$ denotes a learnable matrix of weights, and $\sigma(.)$ is a non-linear activation function. Going back to Equation 4, we can implement a mode of the model that does not incorporate any path information by simply setting $\alpha_{L,k} = 0$ for all $L = 2, \ldots, L_m$ and all $k = 0, \ldots, n-1$. Under this mode, Equation 4 distills to $\mathcal{M} = \alpha_0 A$, which shows that the model is left only with information contained in the adjacency matrix, $A$, to the exclusion of all the path information contained in the tensor, $\mathcal{T}^L$. On the other hand, to implement a mode that incorporates path information, we simply use Equation 4 as it is, without setting $\alpha_{L,k} = 0$. We juxtapose the two modes pictorially in Figures 2a and 2b, for the simple case of $L_m = 2$. In Figure 2a, we see that the diagram includes a cube representing $\mathcal{T}^L$, which encapsulates path information; whereas in Figure 2b, the diagram does not include the cube. An actual Pytorch implementation of the two modes of the model can be found at the URL: `https://github.com/rahmanoladi/kaust_path_project/tree/main/path_project/t_hop`

### 2.3.4   Expressiveness of T-Hop and connections between $\mathcal{B}^L$, $\mathcal{T}^L$, and $A^L$.

Having described the mechanisms underlying the T-Hop model in the previous sub-section, we now turn to study some of its theoretical properties. We will illustrate how its tensor-based representation extends the powered adjacency matrices, which constitute a well known graph representation. Specifically, we will explore theoretical connections amongst: (1) the powered adjacency matrix, $A^L$ ; (2) the T-hop tensors, $\mathcal{B}^L$ and $\mathcal{T}^L$; and (3) the expressiveness of the T-Hop model. We begin with:

**Definition 1** (Cardinality of multiset, $\mathcal{P}_{ij}^L$). *Let $V = \{v_1, \ldots, v_n\}$ be the set of nodes in graph $G$, and let $A_{ij}^L$ denote the $(i, j)$ entry of powered adjacency matrix, $A^L$, so that for two arbitrary nodes, $v_i, v_j \in V$, $A_{ij}^L$ is the number of simple paths of length $L$ connecting $v_i$ and $v_j$. Define $P_q^L = \{v_1^q, v_2^q, \ldots, v_{L+1}^q\}$ as the $q$-th simple path of length $L$ between these nodes, where $v_t^q$ is the t-th node along that path. Let $\mathcal{P}_{ij}^L = \{P_1^L, P_2^L, \ldots, P_{A_{ij}^L}^L\}$ be the multiset containing all such paths of length $L$ between $v_i$ and $v_j$. The cardinality of $\mathcal{P}_{ij}^L$ is denoted as $|\mathcal{P}_{ij}^L|$ and is defined as the total number of nodes within all the paths in $\mathcal{P}_{ij}^L$, counting multiplicities of nodes: $|\mathcal{P}_{ij}^L| = \sum_{q=1}^{A_{ij}^L} |P_q^L| = (L+1)A_{ij}^L$, because $\mathcal{P}_{ij}^L$ contains $A_{ij}^L$ simple paths and each path contains $L+1$ nodes.*

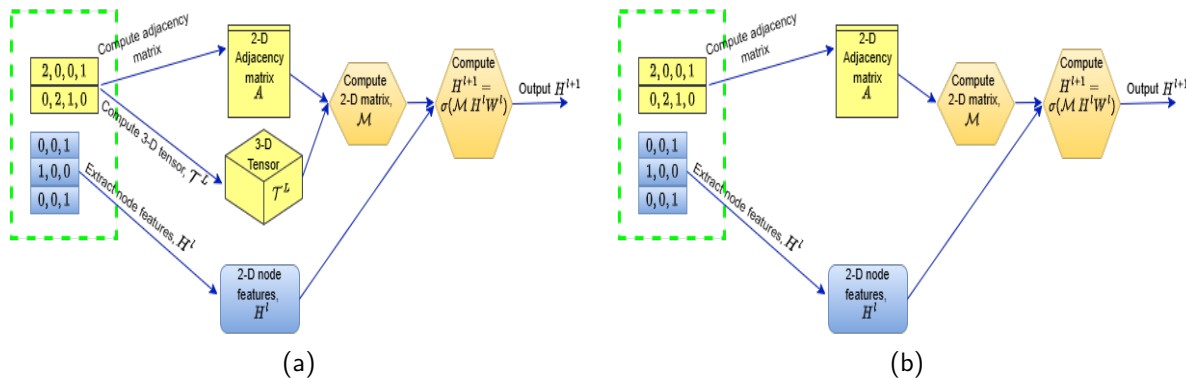

Figure 2: Schematic depiction of the two modes of T-Hop model for the case of $L_m = 2$: (a) using path information contained in tensor $\mathcal{T}^L$ as represented by the cube in the diagram (b) *without* using path information contained in tensor $\mathcal{T}^L$ (so no cube is included in this case.)

Based on the preceding definition, the following lemma holds

**Lemma 1** ($|\mathcal{P}_{ij}^L|$ equals $\sum_k \mathcal{B}_{i,j,k}^L$). *The cardinality of the multiset $\mathcal{P}_{ij}^L$, as defined in Definition 1 above, is equal to $\sum_k \mathcal{B}_{i,j,k}^L$.*

*Proof.* If $\mathcal{P}_{ij}^L$ is the multiset of nodes along all simple paths of length $L$ between two arbitrary nodes, $v_i$ and $v_j$, then for any node $v_k \in \mathcal{P}_{ij}^L$, the multiplicity of $v_k$ —i.e., the number of times $v_k$ occurs in $\mathcal{P}_{ij}^L$—corresponds to the number of simple paths of length $L$ between $v_i$ and $v_j$ that contain $v_k$. But this count equals $\mathcal{B}_{i,j,k}^L$, by the very definition of $\mathcal{B}_{i,j,k}^L$. Hence, the total number of nodes in $\mathcal{P}_{ij}^L$ (counting multiplicities) is the sum of $\mathcal{B}_{i,j,k}^L$ over $k$. This can be written as $|\mathcal{P}_{ij}^L| = \sum_k \mathcal{B}_{i,j,k}^L$. $\square$

The just proven Lemma 1 paves the way to prove the following theorem:

**Theorem 1** ($f_{sum}$ recovers $A^L$ from $\mathcal{T}^L$ ). *Let $f_{sum} : \mathbb{R}^n \to \mathbb{R}$ be the function that sums all components of its input vector $u \in \mathbb{R}^n$. For fixed values of $i$ and $j$, define $t_{ij}^L$ as the $n$-dimensional vector that stretches along the depth axis of $\mathcal{T}^L$ (i.e., $t_{ij}^L = \mathcal{T}_{i,j,:}^L$). Then we have: $f_{sum}(t_{ij}^L) = A_{ij}^L$.*

*Proof.* From Definition 1 we know $|\mathcal{P}_{ij}^L| = (L+1)A_{ij}^L$. But, from Lemma 1, we also know $|\mathcal{P}_{ij}^L| = \sum_k \mathcal{B}_{i,j,k}^L$. Combining both equations, we obtain $\sum_k \frac{\mathcal{B}_{i,j,k}^L}{L+1} = A_{ij}^L$. Now, by definition we know $\mathcal{T}_{i,j,k}^L = \frac{\mathcal{B}_{i,j,k}^L}{L+1}$. Hence, we get $\sum_k \mathcal{T}_{i,j,k}^L = A_{ij}^L$. Finally, it should be clear that $\sum_k \mathcal{T}_{i,j,k}^L$ is tantamount to applying $f_{sum}$ to the vector $t_{ij}^L = \mathcal{T}_{i,j,:}^L$, thus completing the proof. $\square$

Theorem 1 indicates that T-Hop should be more expressive than a model that utilizes the sum of the adjacency matrix, $A$, and the powered adjacency matrices, $A^L$. To see this, note from the theorem that, $A_{ij}^L = \sum_k \mathcal{T}_{i,j,k}^L \implies \sum_{L=2}^{L_m} A_{ij}^L = \sum_{L=2}^{L_m} \sum_k \mathcal{T}_{i,j,k}^L \implies \sum_{L=2}^{L_m} A^L = \sum_{L=2}^{L_m} \sum_k \mathcal{T}_{:,:,k}^L$. Now, if we restrict T-Hop, by setting all learnable parameters in Equation 4 to unity, then $\mathcal{M}$ reduces to: $\mathcal{M} = A + \sum_{L=2}^{L_m} \sum_k \mathcal{T}_{:,:,k}^L = A + \sum_{L=2}^{L_m} A^L$. However, since T-Hop does not impose this restriction, it is therefore expected to be more expressive than a model that uses the sum of the adjacency matrix and the powered adjacency matrices.

### 2.4 Path usefulness measure: a quantifier of the usefulness of incorporating path information in our models.

This subsection introduces the PUM, a metric designed to *quantify the extent to which incorporating path information into a given GNN model enhances its accuracy on a specific dataset family.* Consider a set

of $K$ GNN models, $\mathcal{M} = M_1, M_2, \ldots, M_K$, and a collection of $N$ dataset families, $\mathcal{D} = D_1, D_2, \ldots, D_N$. Each dataset family $D_j$ includes an original dataset along with several synthetic variants generated by adding noise. For each variant $D_j^\gamma \in D_j$, associated with a noise level $\gamma$, we evaluate whether the inclusion of path information improves the model's accuracy. Define an indicator function $\mathcal{F}(M_i, D_j^\gamma)$ such that:

$$\mathcal{F}(M_i, D_j^\gamma) = \begin{cases} 1, & \text{if incorporating path information improves accuracy} \\ 0, \text{otherwise} \end{cases}$$

PUM for a given model $M_i$ and dataset family $D_j$ is then:

$$U_{ij}(M_i, D_j) = \frac{\sum_{D_j^\gamma \in D_j} \mathcal{F}(M_i, D_j^\gamma)}{|D_j|} \tag{6}$$

where $|D_j|$ is the total number of variants in the family. This score represents the fraction of variants within the dataset family for which path information led to improved accuracy. In essence, PUM estimates the probability that incorporating path information enhances model performance for a particular model-dataset pair, providing a normalized measure to compare across datasets and models.

## 3  Experiments

Having established the theoretical foundations, we now explore via experiments and empirical analyses how the use of graph path information impacts the accuracies of GNNs. Our results show that path information does not consistently boost GNN accuracies. Also, our results show an inverse correlation between the benefit of using path information and the level of randomness/irregularities in the graphs. Because incorporating graph path information into GNNs come at an extra computational cost, it follows that one practical importance of our results is that it can be used to inform a more resource-efficient approach, wherein path information is not incorporated when working on graphs with high levels of algorithmic complexity.

### 3.1  Incorporating path information does not yield consistent gains in accuracies on the original six datasets

In this subsection, we report our experimental results assessing how useful including graph path information is for enhancing GNN performance on the six non-synthetic molecular datasets introduced in Section 2.1. We consider three model types discussed in Section 2 and two modes per model—one without path information and one with—resulting in a total of $6 \times 3 \times 2 = 36$ experimental configurations. For each setup, we used the Optuna library (Akiba et al., 2019) to perform hyperparameter sweeps on the validation subset to identify optimal parameters. Using these parameters, we trained each model for 200 epochs on the corresponding training set, then evaluated performance on the test set, recording the relevant metric: RMSE for regression tasks and ROC-AUC for classification tasks. To ensure robustness, each experiment was run three times, and we report the mean and standard deviation of the results. These are shown in Table 2, with the standard deviations placed in parentheses beneath the associated means. Analyzing these results across all configurations, it becomes clear that adding path information does not yield consistent improvements. The additional computational cost therefore is not always justified. Also, this shows that a section of the users of the Graphormer and Mix-Hop models might be able to improve the accuracies of their models simply by switching off the use of path information in those models. Further, this result aligns with prior findings in the literature (Li et al., 2019; Michel et al., 2023; Cong et al., 2023). For example, in the PathNN method (Michel et al., 2023), increasing path length improved results for the NCI1 dataset of the TU Dataset collection (Morris et al., 2020) but failed to do so for PROTEINS and ENZYMES in the same collection. Similarly, in the IGCN paper (Li et al., 2019), experiments on the AWA2 dataset (Xian et al., 2018) showed performance deterioration as the maximum path length parameter $k$ increased from 1 to 3. Moreover, in recent work on dynamic graphs, viz GraphMixer (Cong et al., 2023) reported that a larger receptive field, which corresponds to longer paths, sometimes diminished accuracy.

Table 2: Comparison of models with and without path information across all six original datasets. Path information does not help in all cases.

| DATASET | RMSE (Lower is better) | | | | | | ROC-AUC (Higher is better) | | | | | |
|---|---|---|---|---|---|---|---|---|---|---|---|---|
| | FREESOLV | | ESOL | | LIPOPHILICITY | | BACE | | BBBP | | CLINTOX | |
| | WITHOUT PATH | WITH PATH | WITHOUT PATH | WITH PATH | WITHOUT PATH | WITH PATH | WITHOUT PATH | WITH PATH | WITHOUT PATH | WITH PATH | WITHOUT PATH | WITH PATH |
| Graphormer | | | | | | | | | | | | |
| | 3.55 (0.4539) | **2.91** (0.1556) | 1.20 (0.0762) | **1.01** (0.0018) | **0.91** (0.0147) | 0.99 (0.0357) | **0.782** (0.0002) | 0.735 (0.0283) | 0.686 (0.0172) | **0.709** (0.0089) | 0.602 (0.0128) | **0.779** (0.0065) |
| Mix-Hop | | | | | | | | | | | | |
| | **2.45** (0.0714) | 2.61 (0.1790) | 1.01 (0.0138) | **0.97** (0.0373) | 0.96 (0.0024) | **0.95** (0.0105) | **0.799** (0.0047) | 0.798 (0.0018) | **0.707** (0.0201) | 0.678 (0.0088) | **0.596** (0.0098) | 0.591 (0.0228) |
| T-Hop | | | | | | | | | | | | |
| | 2.87 (0.3241) | **2.73** (0.0385) | **1.00** (0.0330) | 1.00 (0.0367) | **0.78** (0.0210) | 0.88 (0.0210) | **0.852** (0.0087) | 0.746 (0.0346) | **0.649** (0.0032) | 0.623 (0.0363) | **0.785** (0.0079) | 0.752 (0.1003) |

## 3.2 Measuring PUM on the models and dataset families.

Experiments from the previous sub-section showed that incorporating path information does not consistently lead to gains in performance on the three models and on the six original molecular datasets considered. Yet, the experiments did not provide any quantification of the extent to which path information helped. Recall from Section 2.4 that the PUM gives us a criterion for quantifying the extent to which path information helps on a given dataset family, with respect to a given model. Consequently, in this section, we will describe experiments performed to obtain PUMs for the various combinations of (model, dataset family) pairs relevant to this work. Specifically, for our experiments here, we used all the thirty six synthetic and non-synthetic molecular datasets within all the six dataset families listed in the earlier sub-section on datasets (i.e. Section 2.1). However, since we already dealt with the case of the six original non-synthetic datasets in the preceding sub-section, then for those cases, we simply copied the relevant results from the preceding sub-section. The remaining thirty synthetic datasets spawned a total of $30 \times 3 \times 2 = 180$ experimental cases, because we trained three different model types (i.e Graphormer, Mix-Hop and T-hop) at two different modes (i.e. without path information and with path information) on each of the thirty synthetic datasets. Ideally, we should perform a separate hyper-parameter sweep for each of the 180 cases, but that would be too computationally expensive. We reused the hyper-parameters from the original datasets for the synthetic variants to reduce computation. Specifically, suppose $e_n$ represents an experimental case associated with a non-synthetic dataset, and $e_s$ represents an experimental case associated with a synthetic dataset. We applied the hyper-parameters associated with $e_n$ to $e_s$ as long as $e_s$ and $e_n$ share the same model type, dataset family and mode (mode here means whether the experimental case involves the use of path information or not). Based on this, for each of the 180 synthetic-dataset experimental cases, we trained the pertinent model associated with the case for 200 epochs on the training segment of the associated dataset, and used the trained model to obtain scores of a relevant accuracy metric (R.M.S.E for regression datasets and ROC-AUC for classification datasets) on the test segment of the dataset. To address statistical fluctuations, we actually ran each experimental case thrice, and computed the mean and standard deviation of the accuracies of the three runs. We present the means and standard deviations in Table 3, where the standard deviations are shown in parentheses beneath the associated means; for ease of reference, the table also copies results for the cases associated with the non-synthetic datasets from Table 2. In addition to the R.M.S.E and ROC-AUC for each experimental case, the table also displays the PUMs associated with each dataset family and model. Because there are six datasets within each dataset family, the PUMs lie in the range $[\frac{0}{6}, \frac{6}{6}]$. A PUM of $\frac{6}{6}$ indicates complete certainty that the given model always yields better results with path information on the given dataset family, while a PUM of $\frac{0}{6}$ indicates complete certainty that the given model yields better results without path information on the given dataset family. Now, we see from the table that, most of the times, the PUMs are neither equal to $\frac{6}{6}$ nor equal to $\frac{0}{6}$, which shows that, even within a single dataset family, it is difficult to predict whether a given model will benefit from the incorporation of path information. Nonetheless, a closer look at the PUMs for each model type shows that the values associated with the T-Hop model are either very close to $\frac{6}{6}$ ( i.e. $\frac{5}{6}$ on FreeSolv dataset family) or very close to $\frac{0}{6}$ (e.g. $\frac{1}{6}$ on ESOL family and $\frac{0}{6}$ on Lipophilicity family). This shows that, unlike the other models, T-Hop allows us to more easily distinguish between

when path information is helpful versus when it is not helpful. In other words, T-Hop is more effective at highlighting the dichotomy between dataset families that benefit from path information versus those that do not. To quantify this observation, we may compute a *dichotomy score*, $\Phi_i$, for the $i$-th model according to $\Phi_i = \frac{1}{6}\sum_{D_j \in \mathcal{D}} \max\{U_{ij}, 1 - U_{ij}\}$. Here, $U_{ij}$ is PUM of applying model $M_i$ on dataset family $D_j$, and $\mathcal{D}$ denotes our collection of six dataset families. Using this confidence score, we obtained scores of $\frac{29}{36}$, $\frac{24}{36}$ and $\frac{33}{36}$ for the Graphormer, Mix-Hop and T-Hop models respectively. We see that T-Hop very closely approaches a perfect score of 1.

Table 3: PUMs for all model types and dataset families. It is difficult to use the PUMs to predict whether or not path information will boost accuracies.

| MODEL | RMSE (Lower is better) | | | | | | ROC-AUC (Higher is better) | | | | | |
|---|---|---|---|---|---|---|---|---|---|---|---|---|
| | FREESOLV | | ESOL | | LIPOPHILICITY | | BACE | | BBBP | | CLINTOX | |
| | WITHOUT PATH | WITH PATH | WITHOUT PATH | WITH PATH | WITHOUT PATH | WITH PATH | WITHOUT PATH | WITH PATH | WITHOUT PATH | WITH PATH | WITHOUT PATH | WITH PATH |
| Graphormer | | | | | | | | | | | | |
| $\gamma = 0$ | 3.55 | **2.91** | 1.20 | **1.01** | **0.91** | 0.99 | **0.782** | 0.735 | 0.686 | **0.709** | 0.602 | **0.779** |
| | (0.4539) | (0.1556) | (0.0762) | (0.0018) | (0.0147) | (0.0357) | (0.0002) | (0.0283) | (0.0172) | (0.0089) | (0.0128) | (0.0065) |
| $\gamma = 0.1$ | 4.219 | **3.54** | 1.23 | **1.19** | 1.02 | 1.05 | **0.779** | 0.761 | **0.693** | 0.661 | **0.594** | 0.582 |
| | (0.3890) | (0.0853) | (0.1130) | (0.1868) | (0.0474) | (0.0132) | (0.0033) | (0.0340) | (0.0160) | (0.0321) | (0.0094) | (0.0064) |
| $\gamma = 0.2$ | 4.16 | **3.42** | 1.31 | **1.16** | **1.03** | 1.04 | **0.781** | 0.767 | **0.687** | 0.653 | **0.594** | 0.575 |
| | (0.3905) | (0.0912) | (0.0743) | (0.0750) | (0.0559) | (0.0203) | (0.0028) | (0.0283) | (0.0122) | (0.0264) | (0.0100) | (0.0080) |
| $\gamma = 0.3$ | 4.24 | **3.44** | 1.36 | **1.22** | **1.02** | 1.06 | **0.780** | 0.768 | **0.688** | 0.663 | **0.580** | 0.578 |
| | (0.3831) | (0.0721) | (0.0956) | (0.0803) | (0.0262) | (0.0298) | (0.0013) | (0.0267) | (0.0148) | (0.0341) | (0.0141) | (0.0046) |
| $\gamma = 0.4$ | 4.02 | **3.47** | 1.34 | 1.51 | **1.03** | 1.06 | 0.776 | **0.783** | **0.684** | 0.636 | 0.579 | **0.580** |
| | (0.4484) | (0.0434) | (0.1136) | (0.2614) | (0.0312) | (0.0165) | (0.0047) | (0.0089) | (0.0157) | (0.0245) | (0.0240) | (0.0053) |
| $\gamma = 0.5$ | 4.20 | **3.49** | 1.41 | 1.62 | **1.00** | 1.07 | 0.777 | **0.781** | **0.670** | 0.641 | **0.581** | 0.577 |
| | (0.7253) | (0.0349) | (0.2020) | (0.1524) | (0.0391) | (0.0122) | (0.0037) | (0.0059) | (0.0122) | (0.0206) | (0.0176) | (0.0052) |
| PUM | PUM=$\frac{6}{6}$ | | PUM=$\frac{4}{6}$ | | PUM=$\frac{0}{6}$ | | PUM=$\frac{2}{6}$ | | PUM=$\frac{1}{6}$ | | PUM=$\frac{2}{6}$ | |
| Mix-Hop | | | | | | | | | | | | |
| $\gamma = 0$ | **2.45** | 2.61 | 1.01 | **0.97** | 0.96 | **0.95** | **0.799** | 0.798 | **0.707** | 0.678 | **0.596** | 0.591 |
| | (0.0714) | (0.1790) | (0.0138) | (0.0373) | (0.0024) | (0.0105) | (0.0047) | (0.0018) | (0.0201) | (0.0088) | (0.0098) | (0.0228) |
| $\gamma = 0.1\%$ | **2.43** | 2.82 | **1.19** | 1.27 | 1.05 | **1.02** | 0.737 | **0.780** | **0.671** | 0.656 | **0.581** | 0.559 |
| | (0.0838) | (0.2120) | (0.1675) | (0.1040) | (0.0052) | (0.0117) | (0.0195) | (0.0066) | (0.0094) | (0.0215) | (0.0690) | (0.0440) |
| $\gamma = 0.2\%$ | 2.87 | **2.73** | 1.28 | **1.23** | 1.16 | **1.11** | 0.671 | **0.691** | **0.657** | 0.629 | **0.589** | 0.539 |
| | (0.3401) | (0.2040) | (0.1640) | (0.0140) | (0.0377) | (0.0416) | (0.0833) | (0.0734) | (0.0274) | (0.1393) | (0.0148) | (0.0329) |
| $\gamma = 0.3\%$ | **3.16** | 3.23 | 1.48 | **1.41** | 1.23 | 1.23 | 0.674 | **0.752** | 0.602 | **0.614** | **0.578** | 0.555 |
| | (0.2525) | (0.5875) | (0.2173) | (0.0625) | (0.0448) | (0.0763) | (0.0400) | (0.0417) | (0.0198) | (0.0230) | (0.0367) | (0.0263) |
| $\gamma = 0.4\%$ | 3.74 | **3.18** | 1.68 | **1.57** | **1.33** | 1.39 | 0.630 | **0.700** | 0.592 | **0.599** | **0.627** | 0.463 |
| | (0.1838) | (0.2319) | (0.2307) | (0.0652) | (0.0829) | (0.1044) | (0.1005) | (0.0621) | (0.0259) | (0.0303) | (0.0057) | (0.0334) |
| $\gamma = 0.5\%$ | 3.89 | **3.32** | 1.83 | **1.63** | **1.45** | 1.51 | **0.676** | 0.663 | 0.567 | **0.603** | **0.550** | 0.493 |
| | (0.1518) | (0.2235) | (0.3599) | (0.1241) | (0.1081) | (0.1752) | (0.0268) | (0.0423) | (0.0310) | (0.0269) | (0.0619) | (0.0641) |
| PUM | PUM=$\frac{3}{6}$ | | PUM=$\frac{5}{6}$ | | PUM=$\frac{3}{6}$ | | PUM=$\frac{4}{6}$ | | PUM=$\frac{3}{6}$ | | PUM=$\frac{0}{6}$ | |
| T-Hop | | | | | | | | | | | | |
| $\gamma = 0$ | 2.87 | **2.73** | **1.00** | 1.00 | **0.78** | 0.88 | **0.852** | 0.746 | **0.649** | 0.623 | **0.785** | 0.752 |
| | (0.3241) | (0.0385) | (0.0330) | (0.0367) | (0.0210) | (0.0210) | (0.0087) | (0.0346) | (0.0032) | (0.0363) | (0.0079) | (0.1003) |
| $\gamma = 0.1$ | 3.68 | **3.07** | **1.09** | 1.32 | **0.82** | 1.01 | **0.819** | 0.764 | **0.645** | 0.618 | **0.826** | 0.681 |
| | (0.7068) | (0.0387) | (0.0307) | (0.0524) | (0.0052) | (0.0090) | (0.0167) | (0.0136) | (0.0066) | (0.0077) | (0.0061) | (0.0367) |
| $\gamma = 0.2$ | **3.34** | 3.35 | **1.26** | 1.30 | **0.88** | 1.00 | **0.827** | 0.770 | **0.642** | 0.620 | **0.806** | 0.591 |
| | (0.1859) | (0.2472) | (0.1582) | (0.0215) | (0.0383) | (0.0078) | (0.0228) | (0.0033) | (0.01160) | (0.0262) | (0.0023) | (0.0518) |
| $\gamma = 0.3$ | 3.80 | **3.34** | **1.22** | 1.34 | **0.92** | 1.01 | **0.804** | 0.687 | **0.632** | 0.612 | **0.799** | 0.627 |
| | (0.8851) | (0.3717) | (0.0781) | (0.0678) | (0.0241) | (0.0525) | (0.0251) | (0.0718) | (0.0195) | (0.0201) | (0.0151) | (0.0567) |
| $\gamma = 0.4$ | 3.41 | **3.34** | **1.50** | 1.52 | **0.93** | 1.02 | **0.804** | 0.774 | **0.631** | 0.611 | **0.784** | 0.667 |
| | (0.4146) | (0.3023) | (0.3625) | (0.1281) | (0.0286) | (0.0142) | (0.0219) | (0.0359) | (0.0222) | (0.0202) | (0.0212) | (0.0466) |
| $\gamma = 0.5$ | 3.91 | **3.55** | 1.62 | **1.42** | **0.93** | 1.04 | **0.789** | 0.763 | 0.624 | **0.634** | **0.773** | 0.538 |
| | (0.1905) | (0.1852) | (0.1876) | (0.1401) | (0.0333) | (0.0137) | (0.0044) | (0.0223) | (0.0154) | (0.0242) | (0.0095) | (0.0673) |
| PUM | PUM=$\frac{5}{6}$ | | PUM=$\frac{1}{6}$ | | PUM=$\frac{0}{6}$ | | PUM=$\frac{0}{6}$ | | PUM=$\frac{1}{6}$ | | PUM=$\frac{0}{6}$ | |

## 3.3 PUM shows inverse correlation with algorithmic complexity.

A centerpiece of the results from the preceding sub-sections is that it is difficult to predict with certainty when path information can be expected to help for a given model and molecular dataset family. Nonetheless, having a means of making such a prediction would be truly beneficial for deploying GNNs in a more resource-efficient manner. Towards this, we hypothesized that the less random the connectivity pattern of a molecular graph is, the more the path information contained in that graph can be expected to boost the accuracies of GNN models. Since algorithmic complexity is a measure of randomness in graphs (Zenil et al., 2018b), this hypothesis implies that we expect an inverse correlation between the algorithmic complexity of our dataset families, and the PUMs associated with the families. In this sub-section, we will empirically validate

this hypothesis from two perspectives. The first perspective will employ Pearson correlations between PUM and algorithmic complexity, while the second will use clustering analysis. From our work in the previous sub-section, we already have all the required PUMs. Thus, we now turn to how to obtain the algorithmic complexities for our six dataset families. To obtain them, we used BDM (Zenil et al., 2023; 2018a). Here, we point out that all members of the same dataset family share similar values of algorithmic complexity. This is because algorithmic complexity only examines the connectivity patterns of the underlying graphs, without looking at the node and edge features. Hence, to obtain the algorithmic complexity of a dataset family, $D_j$, it suffices to compute the algorithmic complexity of its original non-synthetic member, $D_j^0 \in D_j$. Therefore, for each of the six dataset families, we computed the algorithmic complexity of every single molecular graph in its non-synthetic dataset member, $D_j^0$. We averaged the algorithmic complexity values over all molecular graphs in $D_j^0$. We display the results in the first row of Table 4, wherein we have arranged the results in ascending order from left to right, thereby obtaining ascending order algorithmic complexities (AOAC). Further, Table 4 also contains the PUMs associated with all three models and

Table 4: Exhibition of inverse correlation between AOAC and PUM.

| | DATASETS' AOAC AND MODELS' PUMS | | | | | | CORRELATION BETWEEN AOAC and PUM |
|---|---|---|---|---|---|---|---|
| | FREESOLV | ESOL | BBBP | CLINTOX | LIPOPHILICITY | BACE | |
| **DATASETS' AOAC:** | 105.61 | 216.63 | 488.75 | 510.71 | 577.11 | 717.38 | |
| **MODELS' PUMs:** | | | | | | | |
| Graphormer | $\frac{6}{6}$ | $\frac{4}{6}$ | $\frac{1}{6}$ | $\frac{2}{6}$ | $\frac{0}{6}$ | $\frac{2}{6}$ | -0.84 |
| Mix-Hop | $\frac{3}{6}$ | $\frac{5}{6}$ | $\frac{3}{6}$ | $\frac{0}{6}$ | $\frac{3}{6}$ | $\frac{4}{6}$ | -0.19 |
| T-Hop | $\frac{5}{6}$ | $\frac{1}{6}$ | $\frac{1}{6}$ | $\frac{0}{6}$ | $\frac{0}{6}$ | $\frac{0}{6}$ | -0.81 |
| Across all models | $\frac{14}{18}$ | $\frac{10}{18}$ | $\frac{5}{18}$ | $\frac{2}{18}$ | $\frac{3}{18}$ | $\frac{6}{18}$ | -0.82 |

all six datasets, copied from Table 3. For each dataset family, we also computed the average PUM accross all model types; we placed those average PUMs in the *across all models* row of the table. To validate our hypothesis from a first perspective, we computed the Pearson correlation between the PUMs, listed in Table 4, and the corresponding algorithmic complexities. We display the computed Pearson correlations in the last column of the table. First, according to the correlations in the table, we note that each model showed negative correlation. The graphormer model showed the strongest negative correlation, followed closely by the T-Hop model, with the Mix-Hop model coming third. Moreover, from the table, we also see that the correlation between the average PUMs and their associated algorithmic complexities is also strongly negative (See the *across all models* row of the table). We illustrate these observations graphically in Figure 3. In the figure, the red graph is a plot of average PUMs across all models vs algorithmic complexity. The remaining three plots are for the individual models. All four plots in the figure exhibit negative correlations between PUM and algorithmic complexity. Hence, it becomes obvious that low algorithmic complexity of a molecular dataset family can be taken as a predictor that GNN models are more likely to benefit from path information on the datasets within the family. This result is reasonable, because if the path connectivity pattern in a graph is to be useful for machine learning prediction tasks, then we should want the pattern to be as far as possible from being random/unpredictable. That is, we should want it to have low algorithmic complexity. Further, this result is of practical importance because it leads to a more resource-efficient approach, such that path information is incorporated in GNNs only when working on datasets that have low algorithmic complexities. Clearly, this helps save the extra computational expenses that come with the computation of path information.

To validate our hypothesis from a second perspective, we performed binary clustering analysis. We first clustered the dataset families into two categories based on their associated algorithmic complexities. We then repeated the clustering based on PUMs instead of algorithmic complexities. The results are shown in Table 5. In the table, we display the algorithmic complexities (and PUMs), and place the cluster labels, in parentheses, next to the algorithmic complexities (and PUMs). For the algorithmic-complexities-based clustering, we used 0 to label dataset families corresponding to low algorithmic complexities, and used 1 to label those corresponding to high algorithmic complexities. However, since we hypothesize an inverse relationship between algorithmic complexity and PUM, we inverted the labelling scheme in the case of the PUM-based clustering. For the PUM-based clustering,

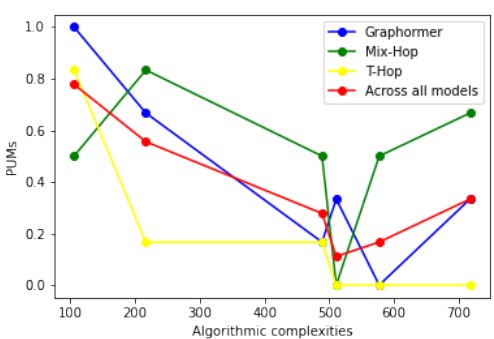

Figure 3: Plots of PUM versus algorithmic complexity

we used 0 to label dataset families corresponding to high PUMs, and used 1 to label those corresponding to low PUMs. Our hypothesis is that the dataset-family clustering produced via the algorithmic complexity values, should match with the dataset-family clustering produced via the PUM. By looking at the *across all models* row in Table 5, we find that the cluster labels there perfectly match the cluster labels in the algorithmic complexities row of the table, thereby validating our hypothesis. For individual models, we found that the Graphormer model also agrees with the hypothesis, with the T-Hop model coming next in agreement, and the Mix-Hop model coming third in agreement. The observed disparities in the levels of agreement with the hypothesis are not far fetched, since the models incorporate path information in different ways. In the table, we also included a column that contains the Silhouette scores for the clustering produced by the three models. We found that the T-Hop model got the highest Silhouette score, which is likely to be due to its heavy and aggressive use of all the available path information contained in the graphs, whereas the graphormer model, for example, only considers path information along shortest paths between graph nodes.

Table 5: Two ways of clustering dataset families: 1) based on algorithmic complexities and 2). based on PUMs. Clusterings produced via the two ways match remarkably.

| | DATASETS' AOAC, MODELS' PUMS AND CLUSTER LABELS IN PARENTHESES | | | | | | SILHOUETTE SCORES |
|---|---|---|---|---|---|---|---|
| | **FREESOLV** | **ESOL** | **BBBP** | **CLINTOX** | **LIPOPHILICITY** | **BACE** | |
| **DATASETS' AOAC & CLUSTER LABELS:** | 105.61 **(0)** | 216.63 **(0)** | 488.75 **(1)** | 510.71 **(1)** | 577.11 **(1)** | 717.38 **(1)** | 0.71 |
| **MODELS' PUMs & CLUSTER LABELS:** | | | | | | | |
| Graphormer | $\frac{6}{6}$ **(0)** | $\frac{4}{6}$ **(0)** | $\frac{1}{6}$ **(1)** | $\frac{2}{6}$ **(1)** | $\frac{0}{6}$ **(1)** | $\frac{2}{6}$ **(1)** | 0.60 |
| Mix-Hop | $\frac{3}{6}$ **(0)** | $\frac{5}{6}$ **(0)** | $\frac{3}{6}$ **(0)** | $\frac{0}{6}$ **(1)** | $\frac{3}{6}$ **(0)** | $\frac{4}{6}$ **(0)** | 0.61 |
| T-Hop | $\frac{5}{6}$ **(0)** | $\frac{1}{6}$ **(1)** | $\frac{1}{6}$ **(1)** | $\frac{0}{6}$ **(1)** | $\frac{0}{6}$ **(1)** | $\frac{0}{6}$ **(1)** | 0.72 |
| Across all models | $\frac{14}{18}$ **(0)** | $\frac{10}{18}$ **(0)** | $\frac{5}{18}$ **(1)** | $\frac{2}{18}$ **(1)** | $\frac{3}{18}$ **(1)** | $\frac{6}{18}$ **(1)** | 0.62 |

## 4 Conclusion

This work advances our understanding of when and why incorporating path information in graph neural networks improves predictive performance on molecular graph datasets. Our empirical analysis uncovers an inverse relationship between dataset algorithmic complexity—estimated using the BDM —and the benefits gained from incorporating path information, as quantified by our proposed PUM. Unlike conventional statistical measures, BDM captures the intrinsic information content arising from a graph's structural ran-

domness (i.e lack of regularity), beyond sheer statistical patterns. By decomposing large graphs into smaller fragments and pre-computing algorithmic complexity estimates on these fragments, BDM provides a refined characterization that reflects the levels of irregularities and randomness, surpassing entropy-based measures in granularity.

Specifically, our results demonstrate that molecular datasets that have lower algorithmic complexity scores—indicating more regular, predictable, and compressible structures—tend to realize significant performance improvements when employing path-aware GNN architectures. Conversely, molecular graphs with higher algorithmic complexity scores, which reflect less structured and more random patterns, show minimal or no benefit from the inclusion of path information, thereby questioning the cost-effectiveness of incorporating such information in these cases. The datasets analyzed primarily involve molecular graphs relevant to cheminformatics and bioinformatics, chosen for their structural diversity and scientific importance in understanding molecular interactions.

We observed the above phenomenon on two popular GNN models, namely Graphormer and Mix-Hop. To further validate it, we introduced T-Hop, a novel tensor-based GNN model explicitly designed to highlight the dichotomy between the use of path information versus its non-use. Amongst all three models studied in this work, T-Hop obtained the highest *dichotomy score* of $\Phi = \frac{33}{36}$, which demonstrates its effectiveness at highlighting the dichotomy between the use of path information versus its non-use.

The clear linkage between a molecular dataset's graph regularity, and the utility of path information provides a pathway toward more resource-efficient deployment of GNNs for molecular datasets. This approach can help reduce resource wastage by not expending the extra computation involved in incorporating path information on datasets that are known to have high algorithmic complexity. This is especially important in large-scale or resource-constrained environments, where high computational complexities are prohibitive. Looking ahead, developing efficient methods to incorporate structure-aware strategies into GNN deployment will be crucial for enabling the creation of adaptive, structure-sensitive models tailored to the specific properties of datasets, thereby advancing resource-efficient and accurate graph-based learning across various application domains. Finally, while the current study was limited to the domain of molecular graphs, we believe that the lessons learnt here will spark researchers in other domains to test our hypothesis on graphs occurring within their domains.

**Acknowledgments**

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
