# OpenReview forum: "Algorithmic Complexity Predicts when Path Information Im- proves Graph Neural Networks Performance on Molecular Graphs"
_TMLR — Rejected by TMLR_

### Review · Reviewer_G2FS · 2025-12-26

**Summary Of Contributions:**

In this paper, the authors investigate the usefulness of incorporating higher order path information in GNNs for molecular graphs. The authors explore this connection by comparing the performances with and without path information of three different GNNs, two well known models from the literature (Graphormer, MixHop) and their own proposed T-hop model, which provides fine grained control over the path information. By correlating their PUM and the graph's algorithmic complexity, via BDM on adjacency matrices, the authors claim that more complex structures benefit less from path information inclusion, and put forward the argument that this principle can be used for downstream applications in order to save computing resources.

**Additional Comments:**

I would like to note that the TMLR review process is double-blind and as such, the authors should not leave traces of their identities in the paper. This aspect is violated in pages 5 and 6, where the linked public GitHub repository is under the name of one of the authors.

**Audience:**

Yes

**Audience Explanation:**

The work done in this paper is relevant to the broader GNN community, in particular to researchers working with molecular graphs. This community is regularly represented in TMLR, which means that the main idea proposed in the paper is relevant and in the scope of the journal.

**Broader Impact Concerns:**

The work does not tackle problems that require a Broader Impact Statement.

**Claims And Evidence:**

No

**Claims Explanation:**

I will start by first listing some strengths of the paper, and then later move towards its weak points, in order to justify my decision on the above. I will expand upon the weaknesses and then report changes (in oder) in the "Requested Changes" section below.

The idea of using the algorithmic complexity of the input graph as a way to study how GNN performance is affected is very interesting and promising. More specifically, the use of graph complexity to characterize how helpful path inclusion has potential, and furthermore this idea can also inspire other similar analyses, making it a potentially impactful contribution.

That being said, there are several issues with the paper that make it not yet ready for publication:
- The biggest issue in my opinion is the experimental protocol used to both define and validate the PUM. Molecular datasets in general have a well defined graph signal, since the features usually encode chemically meaningful information. Continuous and strong perturbations can break both common underlying assumptions in graph learning (homophily, feature-label relationships) and the "chemical realness" of the molecules. These factors are very important as they can completely change the learning dynamics, given that the task labels are defined on the original molecules. Therefore, averaging over different noisy dataset families to get a measure of how useful the path information is could lead to observing effects that are due to ill-defined problems. Considering this issue, Table 2 does indeed provide a helpful initial exploration of the hypothesis the authors put forward, but Table 3 not so much.
- The correlation between PUM and input complexity are very different for Graphformer and Mixhop, and thus it is difficult to reach a conclusion. The paper’s language (“confirmed our hypothesis,” “strong negative correlations”) feels too strong given this evidence base.
- To expand on the above, the relationship between the usefulness of path information will also be dependent on the graph statistics. In fact, the complexity will necessarily be higher for larger graphs, as the number of blocks (and potential repetitions) increases. Normalizing the algorithmic complexities could potentially reveal different correlation.
- Why did the authors only use 6 MoleculeNet datasets and not the complete suite if they wanted to focus on molecular graphs? For example, none of the Quantum Mechanics (QM) datasets are used. On the other hand, from how the introduction is written and also the proposed method, there seems to be no reason to focus only on molecular graphs. This latter statement is not a critique, but on the other hand, if focusing only a specific data type, it would be more informative if more datasets were included.
- The term "path information" is often used but never properly defined. One could assume that it means the inclusion of information at length more than 1 for each node, with the length given by the shortest path metric of the underlying graph. Nevertheless, it should be defined as it is central to the paper. To see where this could lead to doubts, consider Graphormer: when the authors mention that they use it without path information, does this mean they remove the spatial and edge encodings, or just the edge encoding. It is important to have a clear difference in order to correctly understand and interpret results.
- In the theoretical analysis, the authors mention that $A^L_{ij}$ is the number of simple paths of length L between nodes i and j. This is false, as it counts the number of walks of length L, not necessarily simple ones. The authors should take care in accordingly handling their statements.
- Finally, even though the authors state that the proposed analysis has the practical benefit of leading to better efficiency, there are no experiments backing this claim.

Minor issues are:
- Section 2.1 is unnecessarily long considering the target audience of the paper, which are researchers working with or planning to work with GNNs in the molecular domain. It could be significantly shortened.
- The notation in Eq. (1) would be cleaner by just using the index j in the summation, and then defining $r_j$ and $n_j$ after the equation

**Requested Changes:**

-  It could be interesting to average over models with and without path information by expanding over the model family in order to define a new PUM. Naturally, the way each model implements path information is different, but if the claim of relating complexity to path utility is to hold, then it should hold for a broad class of models. Therefore, the authors should consider this alternative. One possibility could be to explore the performance of GCN-based models that use RWSE positional encodings and that don't, as they also include higher-order path information.
- The authors could consider to retune hyperparameters on a small subset of noise levels to report results that are even more fair.
- Add graph statistics for the used datasets, this helps conteztualize the results even more (average degree, number of nodes and edgess ecc)
- Consider providing a clear definition for what "utilizing path information" means, since at a certain point all GNNs use path information (1-hop neighbors require length 1 paths).
- Consider shortening section 2.1
- Since the motivation is resource efficiency, including runtime/memory overhead (or at least complexity scaling) for each “with path” mode would materially strengthen the practical takeaway.

---

### Review · Reviewer_4pTP · 2025-12-28

**Summary Of Contributions:**

This paper investigates the conditions under which incorporating path information benefits Graph Neural Networks (GNNs) for molecular property prediction.
The authors introduce the Path Usefulness Measure (PUM) and demonstrate a correlation between a graph's algorithmic complexity (measured via BDM) and the effectiveness of path data.
By evaluating 36 molecular datasets, they provide a principled framework to decide when path-based features are beneficial or potentially detrimental.

**Additional Comments:**

While the correlation discovered in this study is compelling, the paper currently focuses heavily on establishing the statistical link between algorithmic complexity and the Path Usefulness Measure (PUM).
To further elevate the impact of this work, the authors could expand on the practical implications of these findings.
It would be valuable to discuss how practitioners can use a dataset's complexity score to guide model selection or architectural design, for instance, by automatically toggling path-based features to optimize the balance between computational efficiency and predictive accuracy.
Strengthening this connection between theoretical findings and actionable guidelines would significantly enhance the paper's utility for the broader GNN community.

Furthermore, to ensure that these insights are a fundamental property of molecular graph learning, it is necessary to verify if this relationship holds across a wider range of modern architectures.
While Graphormer and Mix-Hop are solid baselines, testing or discussing these trends in the context of the latest hybrid GNN frameworks (e.g., GraphGPS or MolGPS) would clarify if the PUM-complexity correlation is a universal phenomenon in molecular representation learning.
Strengthening this connection between theoretical findings and diverse state-of-the-art models would make the work much more impactful for the broader GNN community.

**Audience:**

Yes

**Audience Explanation:**

The work is of interest to the TMLR audience, especially those working on GNN expressivity and AI for drug discovery.
Linking information theory (algorithmic complexity) with structural graph learning provides a refreshing and theoretically grounded perspective on a common empirical challenge.

**Broader Impact Concerns:**

No significant concerns are noted.

**Claims And Evidence:**

Yes

**Claims Explanation:**

The claims are generally supported by extensive experiments on Molecule dataset.
The discovery that path information can act as noise in low-complexity graphs is particularly insightful.
To further strengthen the evidence, it would be beneficial to demonstrate that the PUM's predictive power holds across a broader range of contemporary architectures beyond Graphormer and Mix-Hop.

**Requested Changes:**

To ensure the findings are truly universal, I suggest the following additions:

While Graphormer is a strong baseline, testing the PUM framework on more recent state-of-the-art models would add significant value.
GPS (General Powerful Scalable) Graph Transformer / MolGPS: As modern hybrid models (Message Passing + Transformer)

Including standard message-passing GNNs variants would help verify if the observed trends persist in models that don't inherently rely on global path information.

A brief discussion on the computational overhead of BDM vs. the potential performance gains would help practitioners evaluate the trade-off.

---

### Review · Reviewer_7tgY · 2025-12-29

**Summary Of Contributions:**

This paper studies when incorporating path information in GNNs is beneficial for molecular property prediction.
The authors empirically show that path information does not consistently improve performance across molecular datasets, and propose algorithmic complexity as a predictor of when path-aware GNNs are likely to be useful.
They introduce a PUM and conduct experiments on several MoleculeNet datasets using Graphormer, Mix-Hop, and a newly proposed tensor-based model. The main takeaway is that datasets with lower algorithmic complexity tend to benefit more from path information, which the authors argue can guide more resource-efficient model design.

While the problem studied is meaningful and the empirical analysis is fairly extensive, the overall contribution is somewhat incremental.

A major concern is the related work section, which contains very limited coverage of recent literature: only a single paper from 2025 is discussed. This raises the question of whether the literature review is incomplete, or whether the paper is revisiting a research direction whose momentum has already slowed. In either case, the positioning of the work with respect to current trends in graph representation learning is not entirely convincing.

In addition, the presentation quality is a noticeable weakness. Many figures and tables are difficult to read and do not communicate the key insights effectively, which makes it harder to assess the empirical claims. Some plots appear cluttered or under-designed, and the visual narrative could be significantly improved.

Overall, the paper provides an interesting empirical observation linking algorithmic complexity and the usefulness of path information in GNNs, but the novelty is limited, the literature engagement feels insufficient, and the presentation quality detracts from the impact of the results.

**Audience:**

Yes

**Audience Explanation:**

Yes. The paper addresses a relevant question for the GNN community, particularly for researchers interested in understanding when more complex, path-aware architectures are actually beneficial. The empirical observation that path information can sometimes hurt performance, and its connection to dataset properties, may be of interest to practitioners and researchers working on molecular graphs and graph representation learning.

**Claims And Evidence:**

No

**Claims Explanation:**

While the paper presents extensive experimental results, the evidence is not sufficiently convincing or clear to fully support the strength of the claims. The main conclusions rely primarily on correlational analyses over a limited set of molecular datasets, which makes the predictive claims feel suggestive rather than definitive. In addition, several figures and tables do not clearly convey the key trends, reducing the clarity of the supporting evidence. Finally, the limited coverage of recent related work weakens confidence in how well the evidence situates the contribution within the current literature.Yes. The paper addresses a relevant question for the GNN community, particularly for researchers interested in understanding when more complex, path-aware architectures are actually beneficial. The empirical observation that path information can sometimes hurt performance, and its connection to dataset properties, may be of interest to practitioners and researchers working on molecular graphs and graph representation learning.

**Requested Changes:**

- Strengthen and update the related work. The related work section needs a more comprehensive and up-to-date coverage of recent literature (especially from the last 2–3 years). The current discussion includes very limited recent work, which makes it difficult to properly assess novelty and positioning.
- Clarify and appropriately weaken the main claims. The paper should more carefully distinguish between correlational observations and predictive or explanatory claims. In particular, claims that algorithmic complexity predicts when path information is beneficial should be softened or supported with stronger evidence and analysis.
- Improve clarity and presentation of experimental evidence. Key figures and tables need to be redesigned to more clearly convey the main trends and conclusions. As it stands, the presentation weakens the persuasiveness of the empirical results.

---

### Decision · Action_Editor_axs6 · 2026-02-09

**Recommendation:** Reject

**Additional Comments:**

The submission was reviewed by three expert reviewers. Two reviewers found that the claims made in the paper are not supported by accurate, convincing and clear evidence. They requested several changes, such as the inclusion of standard GNNs and more recent models, improved clarity in the presentation of experimental results, a stronger related work section, and an analysis of the runtime and memory overhead of models that utilize path information, just to name a few. The authors did not respond to the reviewers' comments. Two reviewers recommended weak rejection, while one recommended rejection. I agree with the reviewers' assessments and recommend that the authors address the reviewers' comments in a revised version of the manuscript.

**Audience:**

Yes

**Audience Explanation:**

Paths and walks are commonly employed in graph machine learning algorithms to capture structural patterns that are not easily captured by standard message-passing models. Therefore, the findings of this paper will be of interest to some individuals in TMLR's audience.

**Claims And Evidence:**

No

**Claims Explanation:**

The main objective of the paper is to understand when GNN models and Graph Transformers that incorporate path information are beneficial. The study focuses on molecular graphs and reports results on six benchmark datasets. Several key concepts are not clearly defined in the paper. For instance, even though the authors claim to study path information, the two models considered rely on shortest paths and walks, respectively. It should therefore be clarified whether the notion of a path differs from its standard definition. In addition, evaluating only two models is insufficient to draw strong conclusions, and additional architectures should be considered. The claim that algorithmic complexity predicts when path information is beneficial is somewhat strong, as it is not supported by theoretical analysis or strong empirical evidence. Finally, the paper claims that the results may lead to improved efficiency, but no experiments are presented to demonstrate that not incorporating path information results in more resource-efficient models.

**Resubmission Of Major Revision:**

The authors may consider submitting a major revision at a later time.